# The Role of Surgery in Pleural Mesothelioma

**DOI:** 10.3390/cancers16091719

**Published:** 2024-04-28

**Authors:** Moshe Lapidot, Martin Sattler

**Affiliations:** 1Division of Thoracic Surgery, Lung Center and International Mesothelioma Program, Brigham and Women’s Hospital and Harvard Medical School, Boston, MA 02115, USA; 2Department of Thoracic Surgery, Galilee Medical Center, Nahariya 2210001, Israel; 3Department of Medical Oncology, Dana-Farber Cancer Institute, Boston, MA 02215, USA; martin_sattler@dfci.harvard.edu

**Keywords:** pleural mesothelioma, surgery, pleurectomy decortication, extra-pleural pneumonectomy, macroscopic complete resection

## Abstract

**Simple Summary:**

Pleural Mesothelioma is a rare and highly aggressive illness with a grim prognosis. Surgical procedures are crucial in achieving accurate diagnosis and staging. In selected patients, long-term survival is attained when several modalities, including surgery, are combined with the aim of macroscopic complete resection.

**Abstract:**

Surgery plays a central role in the diagnosis, staging, and management of pleural mesothelioma. Achieving an accurate diagnosis through surgical intervention and identifying the specific histologic subtype is crucial for determining the appropriate course of treatment. The histologic subtype guides decisions regarding the use of chemotherapy, immunotherapy, or multimodality treatment. The goal of surgery as part of multimodality treatment is to accomplish macroscopic complete resection with the eradication of grossly visible and palpable disease. Over the past two decades, many medical centers worldwide have shifted from performing extra-pleural pneumonectomy (EPP) to pleurectomy decortication (PD). This transition is motivated by the lower rates of short-term mortality and morbidity associated with PD and similar or even better long-term survival outcomes, compared to EPP. This review aims to outline the role of surgery in diagnosing, staging, and treating patients with pleural mesothelioma.

## 1. Introduction

Pleural Mesothelioma (PM) is an uncommon and highly aggressive form of cancer. In nonresectable patients with performance status 0 to 1 treated with combination chemotherapy, immunotherapy, or chemoimmunotherapy, the median survival is 8.8–20.4 months [1,2]. The 5-year survival rate for all patients with PM is around 5 percent [3]. Over the years, data from both retrospective and prospective cohort studies has shown improvement in overall survival using multimodality treatment, including the incorporation of surgery. Presently, it is well established that surgery plays a crucial role in the diagnosis, staging, and treatment of PM. The guidelines from the National Comprehensive Cancer Network (NCCN), the European Society of Medical Oncology (ESMO), the American Society of Clinical Oncology (ASCO), and the International Mesothelioma Interest Group (IMIG) endorse the use of surgery as part of multimodality treatment in PM. The objectives of surgical intervention in PM encompass obtaining an accurate diagnosis of pleural mesothelioma and its sub-histology, disease staging, participation in multimodality treatment in selected patients, and providing palliative care.

## 2. Diagnosis

The diagnosis of PM and determination of its histology subtype rely on biopsies taken from the pleural tissue. Due to its rarity and similarity in microscopic features it shares with other benign and neoplastic conditions, diagnosing PM might be challenging. Pathological diagnosis primarily depends on histologic evaluation and immunohistochemical characteristics. While pleural fluid cytology can serve as an initial screening test for PM, it is not sensitive enough to make treatment decisions [4]. For obtaining a definitive diagnosis in PM pleural biopsies conducted via thoracoscopy are considered the gold standard but with limitation on the number of ports to obtain multiple tissue samples [5,6]. The European Society of Thoracic Surgeons (ESTS) strongly advocates for obtaining multiple and deep tissue biopsies through thoracoscopy [7], and the ASCO recommends thoracoscopic biopsies for PM patients for whom antineoplastic treatment is planned [4]. The utilization of video assisted thoracoscopy provides a direct visualization of the pleural cavity, mediastinum and diaphragm, to assess the extent of tumor growth [8] and aid in planning definitive surgery. Thoracoscopic biopsies not only contribute to clinical staging but also allow for histologic subtype confirmation and provide material for additional studies such as molecular profiling. The presence of diffuse or nodular pleural thickening on CT scans indicates the likelihood of PM, particularly when the mediastinal pleura is affected. Both the ESMO and NCCN guidelines endorse the utilization of pleural sampling, particularly through thoracoscopy, in patients exhibiting unilateral pleural thickening. The ESMO supports sampling biopsies of at least three distant sites to ensure robust subtyping and grading [9]. The essential role of multiple thoracoscopic biopsies from different parts of the pleural cavity was recently highlighted by our group, based on the discrepancies between preoperative and postoperative histologic subtype diagnosis. In our study, 147 consecutive patients were diagnosed after PD with biphasic histology PM. Among them, only 83 patients exhibited consistency between preoperative biphasic histology and the postoperative histology findings, while 60 patients (40.7%) were initially diagnosed with epithelioid PM, and 4 patients (2.7%) were initially diagnosed with sarcomatoid PM [10]. The data emphasized the need for through sampling to achieve the correct histology subtype. In our clinical approach, we conduct a minimum of three thoracoscopic biopsies from distant areas of the parietal pleura. These biopsies are thorough and extend to include the endothoracic fascia. Although thoracoscopic pleural biopsies that reveal tumor involvement of fat aid pathologists in making a diagnosis, utilizing BAP1/MTAP/CDKTA tests can potentially confirm the diagnosis, even without evident invasion. To address the potential for PM recurrence through incisions and drain tracts we advocate for utilizing a single port for thoracoscopic diagnostic procedures strategically placed on a thoracotomy future incision line that will be excised during PD.

## 3. Staging

The NCCN guidelines support the use of definitive surgery in certain sub-histology. Lymph node assessment is part of the TNM in the AJCC 8th staging system [11], and recommended by either cervical mediastinoscopy or endobronchial ultrasound [12]. Many studies have demonstrated the negative prognostic role of lymph node involvement in pleural mesothelioma [13,14,15]. The ASCO guidelines (Table 1) recommend staging of mediastinal lymph nodes to exclude contralateral or supraclavicular disease (N2 by AJCC 8th), which should be a contraindication to maximal surgical cytoreduction [4]. Interestingly, the role of lymph node involvement as a negative prognostic factor was not demonstrated in non-epithelioid sub-histology mesothelioma, probably due to different biology [10].

## 4. Multimodality Treatment

Studies show that the most favorable long-term results for PM are attained through the combination of several modalities, particularly when surgery and chemotherapy are combined [16,17]. Guidelines endorse the use of surgery for definitive resection with the goal of achieving macroscopic complete resection (MCR). MCR involves the removal of all palpable and visible tumor and can be accomplished through either EPP or PD. Analysis of the SEER database showed that surgery, as a standalone treatment, yields superior overall survival outcomes, compared to no surgery or radiotherapy alone [18]. In a large, matched cohort of 6645 PM patients, cancer-directed surgery was independently linked to enhanced survival rates. Notably, the most significant positive impact was noticed when cancer-directed surgery was incorporated into multimodality treatment [19] 

PM is categorized into three sub-histology—: epithelioid, biphasic, and sarcomatoid [20]. Epithelioid PM accounts for approximately 70%, while biphasic and sarcomatoid PM make up around 15–20% and 10–15% of the cases, respectively. Identifying the specific sub-histology is crucial, due to its significant prognostic implications [4]. In the Surveillance, Epidemiology, and End Results (SEER) database the median survival for patients with epithelioid PM who underwent surgical treatment is 19 months, while biphasic PM has a median survival of 12 months, and sarcomatoid PM a mere 4 months [21]. Generally, while cytoreductive surgery may be considered for epithelioid PM (Table 1), guidelines from the NCCN, ASCO and ESMO advise against offering maximal surgical cytoreduction for patients with histologically confirmed sarcomatoid mesothelioma [4]. Furthermore, the importance of identifying the sub histology is underscored by the significant benefit observed from immunotherapy treatment in the non-epithelioid subgroup in the checkmate 743 trial. This benefit was mainly driven by the poorer response to chemotherapy in this subtype [1]. 

Performing maximal cytoreduction surgery (either lung sparing/PD or non-lung sparing/EPP) should be exclusively considered for patients meeting specific preoperative cardiopulmonary functional criteria, who exhibit no signs of extra thoracic disease, and are deemed appropriate candidates for multimodality treatment (MMT). According to the ASCO guidelines in carefully selected patients with early-stage disease, it is strongly advised to pursue maximal surgical cytoreduction [4]. The rationale behind MMT is based on a recognition that surgical cytoreduction alone is unlikely to achieve an R0 resection, necessitating the administration of chemotherapy and/or radiation. ESMO guidelines draw support for surgical intervention in PM from a variety of retrospective and prospective cohort studies and population and cancer registries, indicating a survival advantage for patients undergoing surgery [22,23]. Factors that suggest lack of benefit from surgery include sarcomatoid histology, contralateral mediastinal or supraclavicular lymph node involvement, extra-thoracic disease, multilevel chest wall infiltration (clinical stage IIIB-IV), or insufficient cardiopulmonary reserve (Table 1). For these cases, systemic or supportive treatments are recommended. The NCCN Panel advices that PD and EPP are viable surgical options that may be appropriate in selected patients to achieve complete gross cytoreduction [5].However, if removal of all visible or palpable tumors (MCR) is unattainable, such as in widespread chest invasion, surgery should be aborted. The panel specifically suggests MCR for epithelioid mesothelioma stage I-IIIA. Additionally surgery might also be an option for individuals with biphasic PM and early-stage disease [5]. Patients with PM who underwent surgery at our institution presented with disease confined to one hemithorax, without involvement of extrathoracic organs, and predominantly exhibited a performance status (PS) of 0–1, along with adequate cardiopulmonary reserve. They underwent evaluation by a specialized multidisciplinary thoracic oncology team, with a specific emphasis on input from mesothelioma surgeons to assess resectability and the potential for achieving macroscopic complete resection. Other centers restricted surgery as part of multimodality treatment to patients with a PS of 0–1, and excluded those with characteristics such as weight loss, thrombocytosis, and leukocytosis, who were directed towards palliative care [24]. The guidelines discuss the role of induction versus adjuvant therapy in MMT but do not offer a definitive recommendation. According to the ASCO guidelines, both induction chemotherapy and sequential chemotherapy after resection are supported without a clear preference for either approach [4]. ESMO states that the sequence of treatment modalities within multimodality treatments involving surgery are not standardized. Typically, Platinum and pemetrexed chemotherapy are administered either before surgery(neoadjuvant) or after surgery (adjuvant), and ongoing research in a phase II clinical trial investigates the optimal timing for this treatment [25]. Notably, in a prospective consecutive cohort of 355 patients who underwent PD as part of multimodality treatment, adjuvant chemotherapy emerged as an independent prognostic factor in a multivariate analysis. However, there was no clear overall survival advantage observed with induction therapy [26]. The efficacy of neoadjuvant chemotherapy was investigated in 257 patients with PM by Voigt and colleagues. Comparing the neoadjuvant therapy group to the immediate resection group, an increased risk in using neoadjuvant for post-resection mortality was noted in both unmatched and propensity-matched cohorts [27]. The authors emphasized the importance of carefully assessing the risks and benefits of induction therapy before offering it to patients with resectable PM.

In the MAPS trial, the addition of bevacizumab (an angiogenesis inhibitor) to chemotherapy demonstrated prolonged survival among nonresectable PM patients [28]. The Checkmate 714 illustrated that combining immunotherapy (Nivolumab + Ipilimumab) increased survival rates in nonresectable PM, particularly in patients with nonepithelioid histology types [1], and the phase 2 PrE0505 trial showed that combining Durvalumab with platinum pemetrexed in nonresectable PM extended overall survival to 20.4 months [2].

However, given multiple complications associated with bevacizumab, such as impaired healing, surgical site bleeding, and wound infection, concerns have arisen regarding its use as neoadjuvant or adjuvant therapy in the multimodality approach. Despite the extended survival outcomes using immunotherapy, either alone or combined with chemotherapy, in nonresectable PM patients, there are currently no published series results addressing its role and effectiveness as part of multimodality treatment.

Due to the technical challenges associated with achieving R0 resection in PM, intraoperative heated chemotherapy (IOHC) has been explored to address microscopic disease and reduce local recurrence rates. Intracavitary hyperthermia enhances the effectiveness of intrapleural chemotherapy by increasing its absorption and activity within tumor cells.

Phase I/II trials investigating EPP with IOHC have indicated that a high dose of 250 mg/m^2^ can be safely administered, with a mortality rate comparable to EPP alone [29,30]. 

Sugarbaker et al. demonstrated survival advantages in a study of 103 in a low-risk PM group receiving cytoreductive surgery and IOHC, compared to those undergoing cytoreductive surgery alone [31].

Phase I/II trials of PD with IOHC using high-dose cisplatin (close to maximal treatment dose) have demonstrated improved survival and prolonged recurrence-free survival, compared to low-dose cisplatin [32]. In our clinical approach, we employ intraoperative heated chemotherapy using cisplatin at 42 °C for a duration of up to 60 min, with measures in place for renal protection. Applying multivariate analysis, we found that administering intraoperative heated chemotherapy (IOHC) to patients who achieved MCR emerged as a significant prognostic factor [26]. 

## 5. Extra Pleural Pneumonectomy

EPP was introduced initially by Irving Sarot in the middle of the 20th century for treating tuberculous infection, and was later adopted by Butchart and colleagues for PM in 1976 [33].EPP involves the en bloc resection of the pleura, lung, ipsilateral diaphragm, pericardium, and mediastinal node sampling. In 1976 Butchart and colleagues reported on 29 consecutive patients who underwent pleuro-pneumonectomy for PM with in-hospital mortality rate of 31%, and only 3 patients surviving for 2 years or more [34]. Due to the high mortality and the limited success, this treatment approach was discontinued for many years [35]. Subsequent advancements in surgical techniques and anesthesia have significantly reduced postoperative complications and mortality rates, leading to improved long-term outcomes. Throughout the 1990′s standardization of surgical technique and perioperative care of EPP has resulted in enhanced outcomes. 

Sugarbaker et al. published in 1999 a significant series comprising 183 PM patients who underwent EPP [36]. The study documented an in-hospital mortality rate of 3.8%, a morbidity rate of 24.5%, a median overall survival of 19 months and a 5-year survival of 15%. Positive prognostic factors identified in this study encompassed epithelioid histological subtype, clear surgical margins and the absence of lymph nodal involvement. In a subsequent series involving 117 patients from the same institution, those who survived more than 3 years exhibited favorable prognostic factors, such as being female, younger in age, having epithelioid histology subtype and normal levels of platelets, white blood cells and hemoglobin [37].

The perioperative morbidity and mortality rates among various series of PM patients undergoing EPP have ranged from 0–82.6% and 0–11.8%, respectively [33,38,39]. The enhanced perioperative survival rates in recent series from high volume centers (approximately 5%) are attributed to improved postoperative care, meticulous patient selection and technical expertise (Table 2). It is crucial to carefully select patients for EPP based on cardiopulmonary assessment and performance status, and the postoperative management necessitates the presence of an experienced intensive care unit team skilled in caring for patients following pneumonectomy. To avoid serious post pneumonectomy complications invasive hemodynamic monitoring is essential for guiding fluid administration. Importantly, centers’ experience was found to be crucial in avoiding postoperative acute respiratory distress syndrome—ARDS [40].

In one of the largest reported series involving 496 patients who underwent EPP, there were 20 postoperative deaths (4%), with the most frequent causes being pulmonary embolism (PE) in six patients, Acute Respiratory Distress Syndrome (ARDS) in four patients, and myocardial infarction (MI) in three patients. Almost exactly two-thirds (sixty six percent) of patients experienced postoperative morbidity [48]. The most common morbidities included atrial fibrillation (44.2%), prolonged intubation (7.9%), vocal cord paralysis (6.7%), deep vein thrombosis (6.4%), patch failure and bleeding (6.1%), ARDS (3.6%), renal failure (2.7%), and empyema (2.4%) [48].

Thromboembolic disease is frequently observed in patients with PM. To mitigate the devastating outcome of PE in a solitary lung, our team conducts noninvasive vascular studies before the operation and on the 7th day after surgery. If a clot is detected in the preoperative study, an inferior vena cava (IVC) filter is placed before surgery. Additionally, we maintain a low threshold for performing CT Angiography scans to screen for PE. 

To reduce the occurrence of late bronchopleural fistula and the onset of empyema we reinforce the bronchial stump with well vascularized tissues. It is crucial to bear in mind that adjuvant radiation, as a component of MMT, may be offered to post pneumonectomy patients. The use of well-vascularized buttress may serve as a protective barrier against bronchial stump dehiscence in such cases. 

Various prognostic scores have been developed to identify patients who might derive benefit from surgery within a multimodality treatment approach. One such score is the multimodality prognostic score, which comprises four variables: histology subtype, pre-induction CRP level, pre-induction tumor volume, and disease progression following chemotherapy [49]. At the Brigham, to classify patients based on the risk of recurrence and mortality, molecular prognostic tests have been integrated with preoperative tumor volume, lymph node status, and histology [50,51].

The comparison between the role of EPP and non-surgical therapy has been examined in only one randomized controlled trial: the Mesothelioma and Radical Surgery (MARS-1) trial. This trial, which was relatively small in scale, revealed poorer survival outcomes in the EPP arm. The MARS trial has sparked debate in the field of PM for several reasons. Criticism has focused on the fact that survival was not the primary endpoint of the study, the small sample size (16 patients in the EPP arm), and the surgical mortality of 18%, which was much higher than the expected surgical mortality in experienced centers (2–5%). Moreover, the trial required a much larger sample size of 670 patients to detect any potential survival benefit from EPP; however, only 50 patients were randomly assigned to treatment.

## 6. Pleurectomy Decortication

Initially PD was considered as a treatment option for PM mainly to palliate symptoms [52]. In 1976, Wanebo and colleagues reported a median survival of 21 months for patients who underwent pleurectomy followed by radiation therapy and adjuvant chemotherapy [53]. PD involves the complete removal of both the parietal and visceral pleura, along with all visible tumor tissue. Preserving the lung distinguishes PD from EPP. Extended pleurectomy decortication (ePD) entails PD, along with the resection of the pericardium and/or ipsilateral diaphragm. 

The extent of pleurectomy can differ based on the tumor burden and surgeon’s preferred type of surgical resection, making it challenging to compare various pleurectomy techniques. In order to reduce procedural variability the International Association for the Study of Lung Cance (IASLC) has introduced a three-level pleurectomy classification system [54].

At Brigham and Women’s hospital, in the early stages of our experience, most patients underwent EPP, with PD being reserved for those unable to tolerate pneumonectomy. Over the past two decades there has been a global increase in the use of PD and ePD for curative purposes. Prior to surgery all PM patients at our institution undergo preoperative assessments, including chest radiograph (CXR), contrast-enhanced computed tomography of the chest (Chest-CT), Magnetic Resonance Imaging of the chest (chest MRI), positron emission tomography-CT scan (PET-CT), spirometry and ventilation-perfusion scan. A multidisciplinary team evaluates the patients for surgical candidacy. Patients diagnosed with mediastinal nodal metastases through cervical mediastinoscopy or endobronchial ultrasound undergo neoadjuvant chemotherapy before being reevaluated for surgery. Reevaluation before surgery was also performedfor patients with ipsilateral extension through vital structures or a chest wall. These patients are restaged prior to surgical consideration and require a treatment response. Our objective of pleurectomy decortication is complete removal of parietal and visceral pleura, along with resection of the diaphragm and/or pericardium if they are found to be macroscopically involved with the tumor or confirmed through frozen section analysis. If necessary, we reconstruct the diaphragm and/or pericardium using permanent patches, which are typically GorTex with pericardial fenestrations. In early stages the resection of either, or both, of these structures may not be necessary to achieve the MCR [55]. 

During PD lymph nodes in the mediastinum, hilum, intercostal and mammary vessels are sampled for pathological staging determination. Previous biopsy sites are excised, and the pleural cavity is irrigated with saline, water and peroxide. Additionally, argon beam painting is carried out on the chest wall and suspected regions of the remaining visceral pleura. Following this, intraoperative heated chemo using Cisplatin at a dosage of 175 mg/m^2^ min is administered and circulated at 42 °C for up to 60 min with renal protection, as previously described by us [29,32]. 

In a significant consecutive series involving 355 patients who underwent PD at Brigham and Women’s Hospital in Boston [26], the observed mortality rates at 30 and 90 days were 3% and 4.6%, respectively. These mortality rates align well with other large series (Table 2). The median survival was 23.2 months with a 5-year survival of 21.2% in the MCR group. Notably, almost 40% of the patients in this series exhibited non epithelioid subtype histology. For patients with epithelioid sub histology and T1 disease, the median and 5-year survival rates were 69.8 months and 54.1%, respectively. The primary complications observed in this series included prolonged air leak (46.5%), deep vein thrombosis (21.1%), atrial fibrillation (13.9%), chyle leak (7.9%), empyema (7.6%), and pneumonia (6.9%). To achieve complete macroscopic resection, attained in 85% of the patients, we conducted an aggressive visceral pleural excision, which contributed to the incidence of prolonged air leaks in our cohort. In 7.6% empyema following surgery occurred and was found to be associated with male sex, use of gortex mesh, and post-operative prolonged air leak [55]. Interestingly the timing of empyema onset differs between PD and EPP. Infante and colleagues [56] noted that most cases of empyema following PD were identified within the first 30 days, while instances of the empyema cases in EPP patients typically manifested later. The discrepancy in timing is probably linked to the differing pathophysiology. Post-EPP empyema is primarily associated with the development of a central bronchopleural fistula, whereas post-PD empyema is predominantly related to a peripheral bronchopleural fistula stemming from perioperative visceral pleura tears and lung injury. The occurrence of postoperative empyema in the cohort of 355 patients who underwent PD was associated with prolonged hospital stays, increased mortality, and notably diminished overall survival rates. In our efforts to mitigate the incidence of prolonged air leaks and reduce postoperative empyema, we have refined our surgical techniques by meticulously addressing all lung lacerations through stapling and/or oversewing all defects with chromic sutures, followed by the application of sealants. 

During the period of May 2016 to August 2018, among patients receiving PD at our institution, the occurrence of venous thromboembolism was 32%. Notably, as many as 33% of patients with deep vein thrombosis were asymptomatic at the time of diagnosis. De Leon et al. suggested that routine surveillance could be beneficial in identifying and managing deep vein thrombosis before it advances to symptomatic or fatal pulmonary embolus [57]. 

In our study involving 355 patients, various factors were identified in a multivariate analysis as being linked to longer overall survival among patients after PD. These prognostic factors included epithelioid histology, female sex, early T stage, lower tumor volume staging, adjuvant chemotherapy, intraoperative heated chemo, and length of stay shorter than 14 days [26]. The overall survival rate observed in this patient group was better than a large cohort of 529 patients from the same institution who underwent EPP by the same surgical team in an earlier period and had a more favorable epithelioid histology [26,45]. Although the proportion of patients with epithelioid histology in each stage group was not significantly different, the median survival for the PD epithelioid subgroup was 31.5 months, compared to 18 months in the EPP group.

Our recent manuscript focuses on 147 females with PM who underwent PD. This cohort of patients achieved a notable median survival exceeding three years, accompanied by low mortality rates at 30 and 90 days. Interestingly, female gender did not emerge as a favorable prognostic factor in non-epithelioid mesothelioma (this paper has been accepted but not published yet).

Available data indicates that PD preserves more lung—and offers comparable overall survival rates, along with reduced postoperative complications and mortality, when compared to EPP in non-randomized research studies [42,58,59,60]. PD, as lung sparing surgery, is associated not just with lower morbidity and mortality, compared to EPP, but also with better quality of life [61,62]. 

In a meta-analysis of 1512 PM patents who underwent PD versus 1391 who underwent EPP, the short term mortality was 2.5 fold higher in the EPP group [63]. The analysis suggests that, given the significantly higher short-term mortality, PD should be preferred if MCR is achievable. 

According to the ESMO guidelines, while EPP was traditionally the preferred surgical procedure, a systematic review involving 1145 patients comparing outcomes between ePD with EPP revealed that that perioperative mortality (2.9% versus 6.8%, *p*—0.02) and morbidity (27.9% versus 62.0%, *p* < 0.001) were significantly lower with ePD, while overall survival rates were comparable [61]. These findings were further supported by a meta-analysis involving 2903 patients who underwent ePD or EPP [63]. As a result, lung-sparing ePD is considered as the first choice surgical procedure, with EPP being proposed in selected patients when performed at specialized high-volume centers. Differences between EPP and ePD are summarized in Appendix A.

The initial findings from a multicenter, randomized trial comparing ePD with no surgery (MARS-2) were recently unveiled at the World Conference on Lung Cancer (WCLC) 2023. In this study 335 patients were randomly assigned to either ePD combined with chemotherapy or chemotherapy alone. The researchers’ analysis indicated that ePD should not be offered to PM due to inferior survival outcomes, a higher occurrence of serious adverse events, and a decline in quality of life in the ePD group. Remarkably, the median survival of 24.8 months observed in the chemotherapy-only group represents the longest reported in randomized intervention trials in PM. The extended survival of patients in the chemotherapy-only arm of MARS2, compared to those in other randomized trials utilizing a cisplatin-based doublet [1,64], hints at a potential selection bias. Criticism was also directed at the trial for its non-standardized pre-randomization phase, which resulted in imbalance between the two groups. Patient characteristics exhibited disparities. For instance, in the surgical arm, there was a twofold increase in sarcomatoid histology type, known for its aggressiveness and grim prognosis, and 83% of patients in this arm had tumors involving the lungs, compared to only 50% in the chemotherapy group. Furthermore, an unusually high 90-day mortality rate was observed in the surgical arm. Although perioperative mortality and morbidity are typically linked to institutional experience, nearly half of the patients in the surgical arm underwent PD in low-volume centers (performing less than 4 PD procedures per year), potentially explaining the elevated 90-day mortality rate of 8.9% (compared to 4.6%, as reported in our paper [59]). The imbalance between the two arms persisted in the post-surgical phase, and despite the trial’s intention to compare chemotherapy alone to PD with chemotherapy, almost 40% of the non-surgical arm received immunotherapy, nearly twice the rate seen in the surgical arm. As the study has yet to be peer reviewed and published, it is advisable to approach these preliminary results with caution. 

## 7. Palliation

Accumulation of pleural fluid in PM that results in breathing difficulties is a common occurrence and often one of the initial symptoms that leads to diagnosis. For patients who are not suitable candidates for extensive surgical cytoreduction, options such as tunneled pleural catheters or pleurodesis (performed via chest tube or thoracoscopy) may be considered. Tunneled pleural catheters are particularly beneficial for promptly relieving recurrent pleural effusions, even in cases when the lung is entrapped [65]. To minimize the risk of cancer cells seeding, these procedures should involve minimal incisions, in terms of both number and size [4]. 

In a randomized controlled trial [66] comparing Video-Assisted Thoracoscopic Surgery (VATS) partial pleurectomy with talc pleurodesis performed through an indwelling chest tube, the primary outcome of overall survival at 1 year showed no significant differences between the two treatment groups. However, VATS partial pleurectomy was associated with a higher rate of complications and longer hospital length of stay. The ESMO guidelines have determined that talc poudrage via thoracoscopy remains the preferred initial procedure for pleurodesis when full lung expansion is achieved. Partial pleurectomy is considered a viable treatment option for patients who are surgically fit but have lungs that are not fully expanded, making them unsuitable candidates for pleurodesis. The comparative effectiveness of VATS-PD versus tunneled pleural catheter in patients with trapped lungs is currently under investigation in a multicenter randomized feasibility phase III trial [67].

## 8. Conclusions

Surgery plays an essential role in the diagnosis, staging, treatment with intent to cure and palliative care for patients with PM. Although there is a lack of properly established clinical trials comparing chemotherapy to surgery-based MMT, the importance of surgery-based MMT protocols for treating mesothelioma is underscored by findings from both retrospective and prospective cohort studies.

The guidelines from the NCCN, ESMO, ASCO, ESTS emphasize the significance of surgical MCR and controlling micro metastatic disease as integral parts of PM MMT protocols. MCR can be accomplished through either PD or EPP. Notably, patients with epithelioid histology, small tumor volume and completion of the entire multimodality treatment tend to have the longest median survival rates. Given the higher mortality and morbidity with EPP over the past two decades many medical centers, including ours, have transitioned from EPP to PD. Several factors might enhance the effectiveness of surgery. These include: 1. Improving case selection by considering favorable prognostic factors, such as histology subtype, low preoperative tumor volume, early T status, female sex with epithelioid histology, and ability to tolerate multimodality treatment [26] 2. Alteration in surgical technique from EPP to PD [59] 3. Decreasing perioperative complications by implementing strategies to minimize occurrences of deep vein thrombosis (DVT) and pulmonary embolism (PE) through routine surveillance measures and early detection of nonclinical events [57]. Additionally, addressing issues like prolonged air leaks and empyema by meticulously repairing all lung lacerations by stapling or over-sewing with sutures [68], and prioritizing primary repair of the diaphragm whenever feasible, avoiding resection and reconstruction with a prosthetic patch [55]. 4. Optimization of operative and postoperative care by performing surgeries in experienced, high-volume mesothelioma centers.

Despite the advent of immunotherapy for PM patients, advances in personalized medicine have been disappointing, compared to other cancers. While surgery will remain an integral modality of PM treatment, it is clear that additional therapeutic options are required, and it will be interesting to see how they will synergize with surgery in a neo-adjuvant or adjuvant setting.

## Figures and Tables

**Table 1 cancers-16-01719-t001:** Indications for surgery (EPP/PD) in PM.

Guidelines	Indications for Surgery(EPP/PD) as Part of Multi-Modality Treatment	Surgery(EPP/PD) Is Not Recommended
NCCN 2024	∗ Clinical stage I–IIIA and Epithelioid histology. ∗ Consider early-stage biphasic	∗ Clinical stage IIIB–IV ∗ Sarcomatoid histology
ESMO 2021	∗ Selected patients in selected centers with such experience (MCR as part of MMT)	∗ Sarcomatoid histology ∗ Contralateral mediastinal or supraclavicular lymph node disease involvement ∗ Extra thoracic disease, multilevel chest wall infiltration ∗ Inadequate cardiopulmonary reserve
ASCO 2018	∗ MCR as part of MMT for early-stage disease with specific cardiopulmonary functional criteria ∗ Transdiaphragmatic disease, multifocal chest wall involvement, or histologically confirmed contralateral mediastinal or supraclavicular lymph node involvement should undergo neoadjuvant treatment before consideration of maximal surgical cytoreduction.	∗ Sarcomatoid histology ∗ Contralateral (N2) or supraclavicular (N2) disease should be a contraindication to maximal surgical cytoreduction
ESTS 2020	∗ Carefully and highly selected PM patients as part of MMT	∗ N2 disease (AJCC 8th edition) ∗ Stage IV ∗ Sarcomatoid or sarcomatoid predominant histology (should not be considered other than in the context of research).

NCCN-National Comprehensive Cancer Network, ESMO-European Society of Medical Oncology, ASCO-American Society of Clinical Oncology, ESTS-European Society of Thoracic Surgeons, MCR-Macroscopic Complete Resection, MMT-Multimodality Treatment, EPP-Extra Pleural Pneumonectomy, PD-Pleurectomy Decortication.

**Table 2 cancers-16-01719-t002:** Perioperative morbidity, mortality, and overall survival in surgical studies with more than 100 PM patients.

Article	Study Design	Type of Surgery/N	Overall Morbidity (%)	Mortality (%)	Median OS (Months)
Sugarbaker et al. (1999) [36]	Retrospective	EPP/183	50% (major 24.5%)	3.8%	19
Rusch et al. (1999) [41]	Prospective	EPP/115	ND	5.25	Stage I-29.9 II-19, III-10.4, IV-8
Flores et al. (2008) [42]	Retrospective	EPP/385	10% (just respiratory complications data)	7.0%	12
Nakas et al. (2014) [43]	Retrospective	EPP/112	ND	ND	19.2 (Excluded 90 d mortality)
Bovolato et al. (2014) [44]	Retrospective	EPP/301	21.6%	4.1% (30 d) 6.9% (90 d)	18.8
Sugarbaker et al. (2014) [45]	Retrospective	EPP/529	ND	5.0% (30 d) 8.0% (90 d)	18
Sharkey et al. (2016) [46]	Retrospective	EPP/133	ND	6.0% (30 d) 13.5% (90 d)	12.9
Kostron et al. (2017) [47]	Retrospective	EPP/141	Major-38%	5.0% (30 d) 10.0% (90 d)	23
Flores et al. (2008) [42]	Retrospective	PD/278	6.4% (just respiratory complications data)	4.0%	16
Nakas et al. (2014) [43]	Retrospective	PD/140	ND	ND	16.2
Burt et al. (2014) [40]	Retrospective	PD/130	Major-3.8%	3.1% (30 d)	ND
Bovolato et al. (2016) [44]	Retrospective	PD/202	10.4%	2.6% (30 d) 6.0% (90 d)	20.5
Lang Lazdunski et al. (2015) [24]	Retrospective	PD/102	29.4%	0 (30 d)	32
Sharkey et al. (2016) [46]	Retrospective	PD/229	ND	3.5% (30 d) 9.2% (90 d)	12.3
Lapidot et al. (2022) [26]	Retrospective	PD/355	72.9% (Major 30.7%)	3.0% (30 d) 4.6% (90 d)	20.7

EPP—Extra Pleural Pneumonectomy, PD—Pleurectomy Decortication, OS—Overall Survival, ND—Not Defined.

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
