# Peer review of "The Role of Surgery in Pleural Mesothelioma"

_cancers, 2024, doi:10.3390/cancers16091719_

Round 1
Reviewer 1 Report
Comments and Suggestions for Authors
The authors wrote the manuscript to describe the role of surgery in malignant pleural mesothelioma (MPM). As a reader, I don't think this review has new scientific insight and inspiration.
1. The authors list guidelines and previous studies, but the discussion of the studies was not deep enough. For example, what kind of MPM patients recruited in these studies and who were benefit from surgery should be described more.
2. MPM can be classified as epitheloid, sarcomatoid and mixed types in histology. The authors should clearly indentified the surgery in different histological types, because the clinical outcomes differ among these histological types.
3. Though this review focus on surgery, we know that MPM is very aggressive and has high recurrent rate and systemic therapies are administrated perioperatively. I think authors should discuss more about the survival benefit of surgery in addition to systemic therapies.
4. There are some advances in systemic treatments in MPM including immunotherapy and anti-angiogenesis agents. It is better to discuss about the role of these systemic treatments in addition to surgery.
5. As above point, the authors should give a future perspective to discuss how to improve the efficacy of surgery in MPM and the advances in therapeutic strategy of surgery.
Comments on the Quality of English Language
I don't have major concern about the qality of English language in this manuscript.
Author Response
Reviewer 3
- The authors list guidelines and previous studies, but the discussion of the studies was not deep enough. For example, what kind of MPM patients recruited in these studies and who were benefit from surgery should be described more.
Recruitment criteria:
*Patients with PM who underwent surgery at our institution presented with disease confined to one hemithorax, without involvement of extrathoracic organs, and predominantly exhibited a performance status (PS) of 0-1, along with adequate cardiopulmonary reserve. They underwent evaluation by a specialized multidisciplinary thoracic oncology team, with a specific emphasis on input from mesothelioma surgeons to assess resectability and the potential for achieving macroscopic complete resection. Other centers restricted surgery as part of multimodality treatment to patients with a PS of 0-1 and excluded those with characteristics such as weight loss, thrombocytosis, and leukocytosis and directed towards palliative care[25].
Lines 143-151
Benefited from surgery:
Various prognostic scores have been developed to identify patients who might derive benefit from surgery within a multimodality treatment approach. One such score is the multimodality prognostic score, which comprises four variables: histology subtype, pre-induction CRP level, pre-induction tumor volume, and disease progression following chemotherapy[53]. At the Brigham, to classify patients based on the risk of recurrence and mortality, molecular prognostic tests have been integrated with preoperative tumor volume, lymph node status, and histology[54][55].
Lines 263-269
Favorable prognostic factors in PM patients who benefited from EPP- Epithelioid histological subtype, clear surgical margins, and the absence of lymph node involvement(Section on EPP lines 216-217).
In those who survived more than 3 years exhibited favorable prognostic factors such as being female, younger in age, having epithelioid histology subtype and normal levels of platelets, white blood cells and hemoglobin. (Section on EPP lines 217-221)
Favorable prognostic factors in patients who underwent PD included epithelioid histology, female sex, early T stage, decreased tumor volume staging, adjuvant chemotherapy, intraoperative heated chemo and length of stay shorter than 14 days(Section PD line 353-356)
Our recent manuscript focuses on 147 females with PM who underwent PD. This cohort of patients achieved a notable median survival exceeding three years, accompanied by low mortality rates at 30 and 90 days. Interestingly, female gender did not emerge as a favorable prognostic factor in non-epithelioid mesothelioma. (This paper has been accepted but not published yet).
Lines 362-366
- MPM can be classified as epitheloid, sarcomatoid and mixed types in histology. The authors should clearly indentified the surgery in different histological types, because the clinical outcomes differ among these histological types.
According to the published guidelines, surgery (PD or EPP) is recommended for stage I-IIIA epithelioid PM or early-stage biphasic PM. Surgery is not recommended for sarcomatoid PM( Table I). The difference in clinical outcomes between the three histological subtypes and recommendations on resection are explained in the multimodality section 109-119).
- Though this review focus on surgery, we know that MPM is very aggressive and has high recurrent rate and systemic therapies are administrated perioperatively. I think authors should discuss more about the survival benefit of surgery in addition to systemic therapies.
PM patients who complete the entire multimodality treatment tend to have the longest median survival rates. In our paper on 355 PM who underwent PD, adjuvant therapy was found to be a favorable prognostic factor(lines 353-356). At that time, a combination of chemotherapy (platinum-based therapy with pemetrexed) was the only FDA-approved therapy. The section on multimodality treatment describes the role of surgery with chemotherapy as neoadjuvant or adjuvant therapy(Multimodality treatment).
- There are some advances in systemic treatments in MPM including immunotherapy and anti-angiogenesis agents. It is better to discuss about the role of these systemic treatments in addition to surgery.
In the MAPS trial, the addition of bevacizumab (an angiogenesis inhibitor) to chemotherapy demonstrated prolonged survival among nonresectable PM patients[29]. The Checkmate 714 illustrated that combining immunotherapy(Nivolumab+Ipilimumab) increased survival rates in nonresectable PM, particularly in patients with nonepithelioid histology types[1] and the phase 2 PrE0505 trial showed that combining Durvalumab with platinum pemetrexed in nonresectable PM extended overall survival to 20.4 months[2].
However, given multiple complications associated with bevacizumab, such as impaired healing,surgical site bleeding, and wound infection, concerns have arisen regarding its use as neoadjuvant or adjuvant therapy in multimodality approach. Despite the exended survival outcomes using immunotherapy, either alone or combined with chemotherapy, in nonresectable PM patients, there are currently no published series results addressing its role and effectiveness as part of multimodality treatment.
Lines 169-181
- As above point, the authors should give a future perspective to discuss how to improve the efficacy of surgery in MPM and the advances in therapeutic strategy of surgery.
Several factors might enhance the effectiveness of surgery:1. Improving case selection based on favourable prognostic factors 2.Alteration in surgical technique from EPP to PD 3. Decreasing perioperative complications by minimizing occurrences of deep vein thrombosis (DVT), pulmonary embolism (PE), empyema, and prolonged air leak and 4. Optimization of operative and postoperative care by performing surgeries in experienced, high-volume mesothelioma centers
Lines 444-449
Reviewer 2 Report
Comments and Suggestions for Authors
I congratulate the authors to their review about the current role of surgery in the management of malignant pleural mesothelioma which was neccessary after the presentation of the MARS-2 trial at the WCLC 2023.
However, I recommend two minor changes and a major correction:
- page 1, line 28: in the phase 2 PrE0505 trial the OS was 20.4 months (I have sent the file to the editor). Therefore, the intervall should be widened.
- page 9, line 306: The MARS-2 trial was presented at the WCLC (World Conference on Lung Cancer) 2023. The IASLC organizes several meetings around the world during the year, therefore this citation is misleading.
- page 9, lines 305-314. The MARS-2 trial is a critically important study. I strictly recommend adding more data from the original presentation and the discussion (I have sent the files to the editor, too).
Author Response
Reviewer 1
- page 1, line 28: in the phase 2 PrE0505 trial the OS was 20.4 months (I have sent the file to the editor). Therefore, the intervall should be widened.
Thank you for your suggestion. The PrEo505 trial, with its extended survival using chemoimmunotherapy, was added
In nonresectable patients with performance status 0 to 1 treated with combination chemotherapy, immunotherapy, or chemoimmunotherapy, the median survival is 8.8-20.4 months [1][2].
Lines 26-29
- page 9, line 306: The MARS-2 trial was presented at the WCLC (World Conference on Lung Cancer) 2023. The IASLC organizes several meetings around the world during the year, therefore this citation is misleading.
Thank you for your correction. We changed to World Conference on Lung Cancer(WCLC) instead of IASLC
The initial findings from a multicenter, randomized trial comparing ePD with no surgery (MARS-2) were recently unveiled at the World Conference on Lung Cancer (WCLC) 2023
Lines 386-387
- page 9, lines 305-314. The MARS-2 trial is a critically important study. I strictly recommend adding more data from the original presentation and the discussion (I have sent the files to the editor, too).
Thank you so much for sending the files. We extended the discussion on the trial
The initial findings from a multicenter, randomized trial comparing ePD with no surgery (MARS-2) were recently unveiled at the World Conference on Lung Cancer (WCLC) 2023. In this study 335 patients were randomly assigned to either ePD combined with chemotherapy or chemotherapy alone. The researchers’ analysis indicated that ePD shouldn’t be offered to PM due to inferior survival outcomes, a higher occurrence of serious adverse events and a decline in quality of life in the ePD group. Remarkably the median survival of 24.8 months observed in the chemotherapy-only group represents the longest reported in randomized intervention trials in PM. The extended survival of patients in the chemotherapy-only arm of MARS2, compared to those in other randomized trials utilizing a cisplatin-based doublet [22] [70] hints at a potential selection bias. Criticism was also directed at the trial for its non-standardized pre-randomization phase which resulted in imbalance between the two groups. Patient characteristics exhibited disparities. For instance, in the surgical arm, there was a twofold increase in sarcomatoid histology type, known for its aggressiveness and grim prognosis, and 83% of patients in this arm had tumors involving the lungs, compared to only 50% in the chemotherapy group. Furthermore, an unusually high 90-day mortality rate was observed in the surgical arm. Although perioperative mortality and morbidity are typically linked to institutional experience, nearly half of the patients in the surgical arm underwent PD in low-volume centers (performing less than 4 PD procedures per year), potentially explaining the elevated 90-day mortality rate of 8.9% (compared to 4.6% as reported in our paper [63]). The imbalance between the two arms persisted in the post-surgical phase, and despite the trial's intention to compare chemotherapy alone to PD with chemotherapy, almost 40% of the non-surgical arm received immunotherapy, nearly twice the rate seen in the surgical arm. As the study has yet to be peer reviewed and published it is advisable to approach these preliminary results with caution. Line 385-409
Reviewer 3 Report
Comments and Suggestions for Authors
Dear Authors and Editor
I read with pleasure this interesting review on the role of surgery in Malignant Pleural Mesothelioma (MPM). Herein authors retraced the history of surgery on MPM by focusing on the differences in terms of surgical and oncological results between EPP and P/D.
Since the nature of the manuscript, a narrative review, there are no major methodological concerns or errors but there are some aspects that should be solved.
In the Diagnosis section authors could improve the discussion by adding advice on where, when and why surgery is recommended for MPM patients (i.e. where pleural biopsies are more diagnostic, how deep should be the biopsy, the role of surgery on calcified pleural thickening). Moreover, the reference of a previous author’s study should include more details to guide the reader.
Table 1 should include kind and extension of surgery which guidelines refer on.
The role of intracavitary treatment such as HITHOC should be discussed. Several authors reported interesting results.
Table 2 reported 2 or more times the experience of the same authors which probably refer to the same patients. Authors could select the more significant study and report eventual comparisons in the same row.
The authors experience should be reported as third part, given the review nature of the manuscript.
Benefit of EPP on P/D (higher rate of MCR, Higher rate of long-term survival) and conversely the beneficial role of a less demolitive surgery (better quality of life, multimodality treatment compliance) could be summarized in a third table.
Author Response
Reviewer 3
Since the nature of the manuscript, a narrative review, there are no major methodological concerns or errors but there are some aspects that should be solved.
In the Diagnosis section authors could improve the discussion by adding advice on where, when and why surgery is recommended for MPM patients (i.e. where pleural biopsies are more diagnostic, how deep should be the biopsy, the role of surgery on calcified pleural thickening). Moreover, the reference of a previous author’s study should include more details to guide the reader.
Thank you for your suggestions. The Diagnosis section now includes the following points:
* In our clinical approach, we conduct a minimum of three thoracoscopic biopsies from distant areas of the parietal pleura. These biopsies are thorough and extend to include the endothoracic fascia. Although thoracoscopic pleural biopsies that reveal tumor involvement of fat aid pathologists in making a diagnosis, utilizing BAP1/MTAP/CDKTA tests can potentially confirm the diagnosis even without evident invasion.
Lines 69-74
*The presence of diffuse or nodular pleural thickening on CT scans indicates the likelihood of PM, particularly when the mediastinal pleura is affected. Both the ESMO and NCCN guidelines endorse the utilization of pleural sampling, particularly through thoracoscopy, in patients exhibiting unilateral pleural thickening.
Lines 56-60
*In our study 147 consecutive patients diagnosed after PD with biphasic histology PM . Among them, only 83 patients exhibited consistency between preoperative biphasic histology and the postoperative histology findings,while 60 patients (40.7%) were initially diagnosed with epithelioid PM, and 4 patients (2.7%) were initially diagnosed with sarcomatoid PM [10]. The data emphasized the need for through sampling to achieve the correct histology subtype.
Lines 64-69
Table 1 should include kind and extension of surgery which guidelines refer on.
We added the definition of surgery(ePD or EPP) to Table 1
The role of intracavitary treatment such as HITHOC should be discussed. Several authors reported interesting results.
Due to the technical challenges associated with achieving R0 resection in PM, intraoperative heated chemotherapy (IOHC) has been explored to address microscopic disease and reduce local recurrence rates. Intracavitary hyperthermia enhances the effectiveness of intrapleural chemotherapy by increasing its absorption and activity within tumor cells.
Phase I/II trials investigating EPP with IOHC have indicated that a high dose of 250mg/m2 can be safely administered with a mortality rate comparable to EPP alone[30][31].
Sugarbaker et al. demonstrated survival advantages in a study of 103 low-risk PM group receiving cytoreductive surgery and IOHC compared to those undergoing cytoreductive surgery alone[32].
Phase I/II trials of PD with IOHC using high-dose cisplatin (close to maximal treatment dose) have demonstrated improved survival and prolonged recurrence-free survival compared to low-dose cisplatin[33]. In our clinical approach, we employ intraoperative heated chemotherapy using cisplatin at at 42°C for a duration of up to 60 minutes, with measures in place for renal protection. Applying multivariate analysis, we found that administering intraoperative heated chemotherapy (IOHC) to patients who achieved MCR emerged as a significant prognostic factor[34].
Lines 182-199
Table 2 reported 2 or more times the experience of the same authors which probably refer to the same patients. Authors could select the more significant study and report eventual comparisons in the same row.
Thank you for your suggestion. Papers using the same patients were deleted in Table 2 and the more significant study was selected
The authors experience should be reported as third part, given the review nature of the manuscript.
Our experience in PD is discussed in the PD section and adds information to significant other manuscripts with at least 100 patients in Table 2
Benefit of EPP on P/D (higher rate of MCR, Higher rate of long-term survival) and conversely the beneficial role of a less demolitive surgery (better quality of life, multimodality treatment compliance) could be summarized in a third table
Thank you for your suggestion
A third table with the benefits of each surgery was added- Supp Table 1
Supp Table 1- Comparison between EPP and ePD
|
|
ePD |
|
EPP |
|
Rate of tumor removal |
|
< |
|
|
Perioperative morbidity rate |
|
< |
|
|
Perioperative mortality rate |
|
< |
|
|
Compliance with multimodality treatment |
|
> |
|
|
Quality of life |
|
> |
|
|
Long term survival |
|
≈ |
|
|
Locoregional recurrence rate |
|
> |
|
EPP-Extra pleural pneumonectomy, ePD-Extended Pleurectomy decortication.
Lines 383-384
Round 2
Reviewer 1 Report
Comments and Suggestions for Authors
I think that authors had response to my comments and expanded the content of this review. It is very good to list the 4 key points to response the comment 5. However, I wish the authors expand more contents according the 4 key points and tell readers how to do it accordingly, and this manuscript will be an excellent review.
Comments on the Quality of English Language
There was only a few error in English, and minor editing is needed.
Author Response
Thank you for your suggestion
We elaborated on the four key points and supplemented them with information and references based on our experience.
Several factors might enhance the effectiveness of surgery. These include: 1. Improving case selection by considering favorable prognostic factors such as histology subtype, low preoperative tumor volume, early T status, female sex with epithelioid histology, and ability to tolerate multimodality treatment[34] 2. Alteration in surgical technique from EPP to PD[63] 3. Decreasing perioperative complications by implementing strategies to minimize occurrences of deep vein thrombosis (DVT) and pulmonary embolism (PE) through routine surveillance measures and early detection of nonclinical events[61]. Additionally, addressing issues like prolonged air leaks and empyema by meticulously repairing all lung lacerations by stapling or over-sewing with sutures and prioritizing primary repair of the diaphragm whenever feasible, avoiding resection and reconstruction with a prosthetic patch[58]. 4. Optimization of operative and postoperative care by performing surgeries in experienced, high-volume mesothelioma centers.
Lines 444-456